# Whole Genome Sequencing Differentiates Presumptive Extended Spectrum Beta-Lactamase Producing *Escherichia coli* along Segments of the One Health Continuum

**DOI:** 10.3390/microorganisms8030448

**Published:** 2020-03-22

**Authors:** Emelia H. Adator, Matthew Walker, Claudia Narvaez-Bravo, Rahat Zaheer, Noriko Goji, Shaun R. Cook, Lisa Tymensen, Sherry J. Hannon, Deirdre Church, Calvin W. Booker, Kingsley Amoako, Celine A. Nadon, Ron Read, Tim A. McAllister

**Affiliations:** 1Department of Food and Human Nutritional Sciences, University of Manitoba, Winnipeg, MB R3T 2N2, Canada; adatore@myumanitoba.ca (E.H.A.); Claudia.NarvaezBravo@umanitoba.ca (C.N.-B.); 2National Microbiology Laboratory, Winnipeg, MB R3E 3R2, Canada; matthew.walker@canada.ca (M.W.); celine.nadon@canada.ca (C.A.N.); 3Lethbridge Research and Development Centre, Lethbridge, AB T1J 4B1, Canada; rahat.zaheer@canada.ca; 4Canadian Food Inspection Agency, National Center for Animal Disease, Lethbridge Laboratory, Lethbridge, AB T1J 3Z4, Canada; noriko.goji@canada.ca (N.G.); kingsley.amoako@canada.ca (K.A.); 5Alberta Agriculture and Forestry, Lethbridge, AB T1J 4V6, Canada; Shaun.Cook@gov.ab.ca (S.R.C.); lisa.tymensen@gov.ab.ca (L.T.); 6Feedlot Health Management Services Ltd., Okotoks, AB T1S 2A2, Canada; sherryh@feedlothealth.com (S.J.H.); Deirdre.Church@albertapubliclabs.ca (D.C.); calvinb@feedlothealth.com (C.W.B.); 7Cumming School of Medicine, University of Calgary, Calgary, AB T2N 4N1 Canada; Ron.Read@albertahealthservices.ca

**Keywords:** One Health, extended spectrum Beta-lactamase producing *E. coli*, antimicrobial resistance, whole genome sequencing, comparative genomics

## Abstract

Antimicrobial resistance (AMR) has important implications for the continued use of antibiotics to control infectious diseases in both beef cattle and humans. AMR along the One Health continuum of the beef production system is largely unknown. Here, whole genomes of presumptive extended-spectrum β-lactamase *E. coli* (ESBL-EC) from cattle feces (*n* = 40), feedlot catch basins (*n* = 42), surrounding streams (*n* = 21), a beef processing plant (*n* = 4), municipal sewage (*n* = 30), and clinical patients (*n* = 25) are described. ESBL-EC were isolated from ceftriaxone selective plates and subcultured on ampicillin selective plates. Agreement of genotype-phenotype prediction of AMR ranged from 93.2% for ampicillin to 100% for neomycin, trimethoprim/sulfamethoxazole, and enrofloxacin resistance. Overall, β-lactam (100%; *bla_EC_, bla_TEM-1_, bla_SHV_, bla_OXA_, bla_CTX-M-_*), tetracycline (90.1%; *tet(A), tet(B)*) and folate synthesis (*sul2*) antimicrobial resistance genes (ARGs) were most prevalent. The ARGs *tet(C), tet(M), tet(32),*
*bla_CTX-M-1_, bla_CTX-M-14_*, *bla_OXA-1_, dfrA18, dfrA19, catB3*, and *catB4* were exclusive to human sources, while *bla_TEM-150_*, *bla_SHV-11–12_*_,_
*dfrA12, cmlA1*, and *cmlA5* were exclusive to beef cattle sources. Frequently encountered virulence factors across all sources included adhesion and type II and III secretion systems, while IncFIB(AP001918) and IncFII plasmids were also common. Specificity and prevalence of ARGs between cattle-sourced and human-sourced presumptive ESBL-EC likely reflect differences in antimicrobial use in cattle and humans. Comparative genomics revealed phylogenetically distinct clusters for isolates from human vs. cattle sources, implying that human infections caused by ESBL-EC in this region might not originate from beef production sources.

## 1. Introduction

Infections caused by extended-spectrum β-lactamase (ESBL)-producing *Enterobacteriaceae* have been linked to their dissemination through both food and the environment [1,2,3]. ESBL-producing bacteria, including *E. coli* are frequently resistant to multiple antimicrobials including 3rd and 4th generation cephalosporins in addition to quinolones and aminoglycosides [1,2,4]. These bacteria have been designated priority group 1 pathogens for the development of alternative antimicrobial therapies by the World Health Organization [5] as they can cause serious urinary tract [6,7] and bloodstream [8,9,10] infections, occasionally resulting in mortalities [11].

Genetic determinants of ESBL resistance are commonly associated with mobile genetic elements such as plasmids, transposons, and integrative conjugative elements (ICEs), which facilitate horizontal gene transfer among bacteria [4,12,13]. Plasmids have been frequently identified in *E. coli* [14], while ICEs appear to be far less common than in other pathogenic bacteria such as *Staphylococcus aureus* and *Vibrio cholera* [15,16].

Antimicrobials are used in livestock production for both prophylactic and therapeutic purposes, practices that can promote antimicrobial resistance (AMR) [17,18,19]. A number of reports have characterized AMR in isolated segments of the beef production system, often employing traditional methods of AMR surveillance [19,20,21]. Conventional AMR susceptibility tests have been instrumental in detecting and monitoring AMR status and emergence, but has limitations with regard to the extent that information on phylogeny, mobile genetic elements (MGEs), or virulence traits can contribute to epidemiological investigations [22,23].

Whole genome sequencing (WGS) can make significant contributions to AMR surveillance and in-depth analyses of ESBL-producing *E. coli* (ESBL-EC) of varying origin so as to identify potential “One Health” linkages [24,25]. This approach can help elucidate the genetic relationship among isolates [23] and to identify their origin within the food production chain [26,27]. Moreover, increasing accessibility and decreasing cost of high throughput WGS has made it the method of choice for routine surveillance and outbreak investigation [28].

An integrated One Health approach is needed to consider animal, food, environment, and human contributions to AMR ecology [29,30]. However, there is a dearth of information about the contemporaneous occurrence of AMR along the One Health continuum [31]. Considering the impact of ESBL infections and the prioritization of these bacterial agents, the present study specifically investigated presumptive ESBL-EC from a One Health continuum linked to the beef production system in southern Alberta, Canada. The objective was to characterize presumptive ESBL-EC with respect to phenotypic and genotypic AMR; phylogeny; and the prevalence of plasmids, ICE and virulence factors among isolates. We hypothesized that, although presumptive ESBL-EC from human and cattle sources along the One Health continuum may harbor some similar ARGs, MGEs, and virulence factors, the accessory genomes of isolates from beef cattle production environments and human sources would be genetically distinct.

## 2. Materials and Methods

### 2.1. Study Area, Sampled Sites, and E. coli Isolation

Detailed description of feedlot cattle feces, catch basin, surface streams as well as processing plant, waste water treatment plant, and clinical sources from which presumptive ESBL-EC were collected are published elsewhere [32,33,34]. Cattle feces (CFeces) and liquid samples from catch basins (CBasins) were collected from four feedlots (A, B, C, and D) monthly over 2 years (March 2014 – April 2016). Sampling of feces from cattle was conducted according to the protocol approved by the University of Calgary Animal Care Committee (AC14–0029). In feedlot C, liquid samples were also collected from a surrounding stream (SStreams). Sewage samples were collected from treatment plants located upstream (Calgary) and downstream (Medicine Hat) of the major feedlot cattle production region in Alberta, while human clinical isolates were randomly selected from anonymous individual patients for the same time period from the Division of Medical Microbiology, Calgary Laboratory Services (CLS) biorepository. This laboratory services about 1.5 million people in the City of Calgary and surrounding rural area. Beef processing plant (BProcessing) samples were obtained from ground beef and retail meat and from swabs from hides, beef trimmings, washed carcasses, and conveyers following the procedures of Gills and Jones [35].

Samples from CFeces and BProcessing were individually enriched overnight at 37 °C in *E. coli* broth supplemented with 2 µg/mL cefotaxime (MilliporeSigma, Burlington, MA, USA) to increase the likelihood of isolating presumptive ESBL from cattle and their environments, while enrichment was not necessary for human sources. Presumptive ESBL-EC from each feedlot sample were isolated by adding 0.5 g cattle feces to 4.5 mL of *E. coli* broth containing 2 µg/mL cefotaxime) for overnight enrichment at 37 °C (Cameron-Veas et al., 2015; Ibrahim et al., 2016). Each resultant culture was mixed, and a saturated swab was used to inoculate MacConkey plates supplemented with 1 µg/mL of ceftriaxone (MilliporeSigma) and incubated overnight. A maximum of three distinct red/magenta lactose-fermenting colonies were randomly isolated and subcultured onto tryptic soy agar (TSA)-ampicillin (32 µg/mL) (MilliporeSigma) for purification and subsequent archiving at −80 °C. Presumptive ESBL-EC from CBasins and SStreams were isolated via membrane filtration onto MacConky plates with cefatriaxone (1 µg/mL) using the US EPA Method 1603 [32,33]. The same procedure without enrichment or the use of selective plates was used to isolate generic *E. coli*. All isolates were archived in Brain Heart Infusion (BHI) broth containing 15% glycerol and stored at −80˚C. *E. coli* isolated from selective plates with or without enrichment were described as presumptive ESBL-EC and were the only isolates selected for whole genome sequencing.

### 2.2. Antimicrobial Susceptibility Tests

Briefly, resistance of presumptive ESBL-EC (*n* = 705) to ampicillin, amoxicillin/clavulanic acid, ceftiofor, ceftazadime, streptomycin, neomycin, ceftazidime/clavulanic acid, oxytetracycline, florfenicol, sulfisoxazole, and trimethoprim/sulfamethoxazole was evaluated using the Kirby–Bauer disk diffusion method according to the Clinical and Laboratory Standards Institute [36]. Zone diameters of inhibition were documented with BIOMIC V3 Microbiology System (Giles Scientific, Santa Barbara, CA, USA) and resistant, intermediate, and susceptible frequencies computed as described elsewhere [36]. On the basis of ESBL phenotype, AMR phenotype, isolate source, and sampling site, a representative subset of presumptive ESBL-EC isolates (*n* = 162) was selected for WGS from (i) beef production sources: CFeces *n* = 40; CBasin, *n* = 42; SStreams *n* = 21; BProcessing *n* = 4, and from (ii) humans sources: sewage MSewage *n* = 30 and clinical CHumans *n* = 25.

### 2.3. Whole Genome Sequencing and Assembly

From glycerol stocks, presumptive ESBL-EC were cultured on sheep blood agar overnight at 37 °C and a single colony was selected and subcultured on the same media. A single *E. coli* colony was selected and suspended (~2 × 10^9^ cells/mL) in 10 mM Tris and 1 mM EDTA (TE, pH 8.0) at an OD_600_ of 2. From each suspension, 1 mL was dispensed into a microcentrifuge tube and centrifuged at 14,000× *g* for 2 min to form a cell pellet for DNA extraction. Genomic DNA were extracted using the DNeasy Blood and Tissue kit (QIAGEN, Hilden, Germany) according to the manufacturer’s instructions. DNA quality was assessed using Nanodrop 2000 (NanoDrop Technologies, Inc., Wilmington, DE, USA) and was considered acceptable with an A_260/280_ ~1.8–2.0 [37,38]. Genomic libraries were constructed using the Illumina Nextera XT DNA sample preparation kit with Nextera XT index kit for multiplexing according to the manufacturer’s instructions (Illumina Inc., San Diego, CA, USA). Libraries were then sequenced on the MiSeq Whole Genome sequencing platform (2 × 300 bp paired-end, Illumina Inc.) to obtain an average coverage depth of >100× and a read retention of >98%. The adaptor sequences were cropped with Trimmomatic 0.36 [39] using the following conditions: phred33, LEADING:3 TRAILING:3 SLIDINGWINDOW:4:15 MINLEN:36. Sequences were assembled using SPAdes 3.10.1 [40] at default parameters with modifications: K-mer values (21,33,55,77,99,127), careful correction, length for average coverage calculation (5000), coverage cutoff ratio (0.33), repeat cutoff ratio (1.75), and contig length cutoff (1000). Sequences were annotated with PROKKA [41] at default settings with modifications: BLASTP e-value cutoff for annotation (1e-06), minimum contig size (200), locus tag prefix (EC), locus tag counter increment (1), GFF version (3), searching for ncRNAs with Infernal+Rfam (false) enabled, strain name (blank), species (*coli*), genus (*Escherichia*), kingdom (bacteria), genetic code (11), and plasmid (blank). Downstream WGS analyses were implemented using appropriate programs within a Galaxy pipeline (0.3.0) [42] for identification of resistance genes, plasmids, and virulence genes as detailed in Section 2.4.

### 2.4. Whole Genome Sequence Analyses

Pan genome analyses was conducted using Roary [43], followed by AMR mechanism identification using the National Center for Biotechnology Information (NCBI) and ResFinder databases [44]. ARGs were designated as present if sequence identity was ≥90%. Agreement between ARGs and AMR phenotypes were examined as described by McDermont et al. [24], where resistance or susceptibility to a given antimicrobial was compared with the presence or absence of a known corresponding resistance gene(s) and/or structural gene mutation(s). WGS sensitivity was computed as the number of isolates that harbored resistant determinants divided by the total number of isolates exhibiting clinical resistance, with the phenotypic resistance as the standard outcome. Specificity was calculated as the number of isolates that did not harbor genetic determinants divided by the total number of phenotypic susceptible isolates [24,45]. Separate binary logistic regression [46] was used to examine a subset of cattle versus human isolates (CFeces and CBasins vs. CHumans and MSewage) for differences in ARG prevalence. The few isolates obtained from the BProcessing environment did not allow for them to be included in more extensive data analyses.

Based on single nucleotide polymorphisms (SNPs) in the core genes, phylotyping was performed using SNVPhyl [47] to generate a maximum likelihood tree with the NCBI reference strain *E. coli* str. K-12 substr. MG1655. The maximum likelihood tree was generated for the entire 162 presumptive ESBL *E. coli* isolates as well as a subset of 108 isolates which harbored *bla_TEM_, bla_CTX-M_, bla_OXA_*, or *bla_SHV_* and were designated as true ESBLs. The reference strain *E. coli* MG1655 was used as it has been widely used by others [48]. The tree was subsequently visualized using iTOL [49]. Whole genome multi-locus sequence typing (wgMLST) and a minimum span tree was also generated using BioNumerics (BN v7.6) (BioNumerics, Applied Maths, Keistraat, Belgium) with default parameters for categorical data. Multi-locus sequence typing (MLST) was achieved with PubMLST *E. coli*#1 (Achtman) scheme [50], while virulence factors were identified from the Virulence Factors Data Base (VFDB) [51]. To investigate the prevalence of ICEs in *E. coli,* BLAST homology searches were conducted against 1032 ICE sequences obtained from ICEberg database 2.0 [15]. In silico *E. coli* subtyping was performed based on outer membrane lipopolysaccharide (O) and flagellar (H) surface antigens using an in-house ECTyper and EcOH [52,53] to designate isolates as either O or H groups.

A set of presumptive ESBL-ECs randomly selected from each source of the beef production system that identified as non-ESBL-EC by WGS (total *n* = 23 not carrying any *bla_TEM_, bla_CTX-M_, bla_OXA_*, and *bla_SHV_*) and those identified as true ESBL-EC (*n* = 24 carrying *bla_TEM_, bla_CTX-M_, bla_OXA_,* and *bla_SHV_*) were compared using a GView BLAST atlas [54] to elucidate possible similarities or differences in genomes of the two groups using *E. coli* MG1655 as reference. The draft genome sequences of the 162 *E. coli* isolates have been deposited to GenBank under Bio project PRJNA556083.

## 3. Results

### 3.1. Occurrence of Phenotypic Antimicrobial Resistance

Of the phenotypic resistances tested, oxytetracycline and ampicillin resistance was most common, while neomycin resistance was the least (Figure 1; Appendix A). Although the CLSI minimum inhibitory concentration (MIC) ampicillin break-point was used for isolating presumptive ESBL-EC, not all *E. coli* were resistant to ampicillin based on subsequent disk diffusion susceptibility tests. Differences may have resulted from resistance gene silencing, loss of plasmids, or perhaps differences between MIC and disk diffusion assays. Among the isolates collected, 1.9% were not resistant to any of the antimicrobials, 5.6% were resistant to only one class, while 69.1% were resistant to five or more classes. Overall, a total of 92.6% of *E. coli* strains showed MDR in terms of resistance to three or more classes of antimicrobials. It is worthy to note that, despite enrichment, a limited number of presumptive ESBL-EC isolates were obtained from BProcessing environments (*n* = 4), an attestation to the sanitary procedures within the plant.

### 3.2. Antimicrobial Resistance Genes Prevalence

*Tetracyclines*: A total of 77 unique ARGs encoding diverse antimicrobial resistance were detected. The tetracycline determinants *tet(A)* and *tet(B)* were detected in both cattle and human sources, whereas *tet(C)* and *tet(M)* were detected in cattle sources, but not MSewage nor CHumans. In contrast, *tet(32)* was exclusive to isolates associated with human sources (Figure 2; Appendix A).

*Beta-lactams:* β-lactam *bla_EC-8_, bla_EC-15_, bla_TEM-1_, bla_CTX-M-15_, bla_CTX-M-55_, bla_EC-18_,* and *bla_CMY-2_* were present in all CBasins, CFeces, SStreams, CHumans, and MSewage isolates, whereas *bla_TEM-150_* was detected only in cattle-sources (CBasins, CFeces, and SStreams). In contrast, *bla_CTX-M-1_, bla_CTX-M-14_*, and *bla_OXA-1_* were detected only in CHumans and MSewage, and *bla_CTX-M-27_* and *bla_CMY-42_* were exclusive to human clinical isolates. Additionally, *bla_LAP-2_* occurred in isolates from CBasins and CFeces, with *bla_SHV-11_* and *bla_SHV-12_* occurring exclusively in isolates from CFeces.

*Folate pathway inhibitors:* For folate synthesis inhibitors, *sul2* was ubiquitous but *sul1* was not detected in BProcessing isolates, whereas *sul3* was not found in either BProcessing or SStreams isolates. *dfrA1* was present in MSewage, CBasins, and CFeces, while *dfrA12* was unique to CBasins, CFeces, and SStreams. *dfrA7* was unique to CBasins and *dfrA18,* and *dfrA19* was unique to CHumans isolates. *DfrA14, dfrA17*, and *dfrA27* were present in all CBasins, CFeces, SStreams, CHumans, and MSewage isolates.

*Aminoglycosides:* Aminoglycoside resistance genes also revealed the ubiquity of *aph(3’’)-Ib* and *aph(6)-Id.* The genes *a**ph(3’)-Ia* and *aadA1* were identified in CHumans, MSewage, CBasins, and CFeces isolates, whereas the *aph(3’)-IIa* and *aph(6)-Ic* were only present in CFeces and MSewage isolates. *A**ac(3)-IIa, aac(3)-IId*, and *Sat1(A7J11)* occurred in CHumans and MSewage, whereas the latter two determinants were also identified in CFeces and CBasins isolates. Isolates from MSewage, CBasins, and CFeces possessed *aadA16* and *armA*, and *aac(3)-Via* and *aadA22*, respectively. *AadA2* were detected in CBasins and CFeces, while *aac(6’)Ib-cr* was only found in isolates originating from humans or sewage.

*Phenicols:* Phenicol resistance determinants *catB3* and *catB4* were only found in CHumans and MSewage isolates, whereas *cmlA1* and *cmlA5* were unique to CBasin and CFeces isolates. In contrast, *floR* was found in isolates collected from all environments.

*Quinolones*: The plasmid-borne quinolone-resistant gene *qnrS1* was also ubiquitous, while *qnrB2* was only found in CHumans isolates (Figure 2; Appendix A). Aside from plasmid-borne quinolone resistant *qnr* and *aac(6’)Ib-cr* genes, evaluation for mutations in the quinolone resistance-determining regions (QRDRs) of *gyrA, gyrB, parC, and parE* aligned with frequently reported mutations (*gyrA*: S83L, S83A, D87N, and D87Y; g*yrB*: E185D; *parC*: S80I and E84G; *parE*: S458T and I529L [55,56]), but additional mutations were found that may also have also conferred quinolone resistance (Appendix A). All BProcessing isolates harbored only *tet(C)*, *sul2*, *bla_EC-18_*, *bla_CMY-2_*, *aph(3’’)-Ib*, *aph(6)-Id*, and *floR* genes.

*Overall:* In cattle- and human-derived isolates, β-lactam (100% in both sources) and tetracycline genes (91.6% and 87.3%) were prevalent, with quinolone genes being the least prevalent (20.6% and 10.9%) (Appendix A and S2; Appendix A). When cattle (CFeces and CBasins) and human (CHumans and MSewage) isolates were compared, differences in prevalence were observed. The prevalence of 13 ARGs (*sul1, sul2, bla_EC_, bla_EC-8_, bla_EC-15_*, *bla_EC-18_, bla_CTX-M-15_, bla_CMY-2_, aph(3’’)-Ib, aph(6)-Id, floR, bla_TEM-1_*, and *bla_CTX-M-55_*) differed (*p* < 0.05; Appendix A) between isolates originating from cattle vs. human sources and were generally more prevalent in human isolates, with the exception of *bla_TEM-1_* and *bla_CTX-M-55_*.

*Misc:* Determinants for resistance to rifamycin (*arr-2* and *arr-3*), lincosamides (*lnu(F)*), bleomycin (*ble*), streptothricin (*sat1*), and quaternary ammonium compounds (QAC), efflux small multidrug resistance transporter genes (*qacG2, qacEΔ1* and *qacL*) were also detected. Rifamycin resistance was found in CFeces (7.5%) and MSewage (10%) isolates; bleomycin was found only in to MSewage (3.3%) isolates; and streptothricin in CBasins (2.9%), CHumans (4%), and MSewage (10%) isolates, while lincosamide determinants were present only in CFeces (1.9%) and MSewage (3.3) isolates. The QAC efflux genes were less frequent in CFeces (17.5%), CBasin (19.0%), and SStreams (28.6%) than in MSewage (50%) and CHuman (48%) isolates (Figure 2; Appendix A). Categorically *mph(E), msr(E), erm(B), arr-3*, and *ble* were only found in human-sourced isolates, while *qacG2* were exclusively associated with cattle-sourced isolates.

### 3.3. Pan-Genome Analysis and Compairson of AMR Determinants as Identified by WGS to the Occurrence of Phenotypic Reisistance

*E. coli* genomes ranged from 4.6–5.7 Mbp (Appendix A). Analyses of the 162 isolates identified a total of 24,338 genes, with 2763 core genes in all isolates, 327 soft core genes present in 95%–99% of isolates, 2789 shell genes present in 15%–95% of isolates, and 18,459 cloud genes present in less than 15% of isolates.

Sensitivity for AMR prediction by WGS compared to clinical resistance phenotype varied per antimicrobial class and antimicrobial agent, with ranges between 93.2% (ampicillin) and 100% (neomycin, trimethoprim/sulfamethoxazole, and enrofloxacin), while specificity ranged between 1.1% (enrofloxacin) to 100% (ampicillin) (Table 1; Appendix A). WGS-positive predictive values also ranged between 7.7% (neomycin) and 100% (ampicillin), while negative predictive values ranged between 61.5% (ampicillin) and 100% (neomycin, trimethoprim/sulfamethoxazole, and enrofloxacin). WGS revealed that 90.7% of all the presumptive ESBL-EC were true ESBL-producers based on the presence of bla_SHV_, bla_TEM_, bla_OXA_, bla_CTXM_, or bla_EC-13–15–18–19,_ with gene prevalence not differing by source (*p* = 0.89). In retrospect, both PCR and WGS identified comparable levels of true ESBL-EC 68.5% and 66.7% respectively, based on the presence of *bla_SHV_, bla_TEM_, bla_OXA_,* or *bla_CTXM_* genes.

### 3.4. Phylogenetics, MLST Cluster, and Serogroups Analysis

Evaluation of the genetic relatedness of the beef/cattle-associated and human-associated isolates using a maximum likelihood phylogenetic tree showed that 71.64% of the core genome of isolates from these sources was shared. Limited phylogenetic intermixing among isolates from segments along the One Health continuum was observed (Figure 3). Single nucleotide variant phylogenomics revealed that, in general, human-sourced ESBL-EC comprising CHumans and MSewage formed one major cluster within which a number of distinct clades and sub-clades occurred. The majority of CHumans and MSewage isolates were closely related, although about 7.5% of all beef/cattle-associated isolates (*n* = 107) were in this cluster. Interestingly, 92.5% of the isolates from CFeces, CBasins, and SStreams formed a separate cluster of mixed clades with only a few MSewage (*n* = 9) and CHumans (*n* = 2) isolates in this clade. Isolates from CBasins and CFeces often clustered together. Although the few BProcessing isolates formed a distinct sub-clade from CBasins and SStreams, they also clustered separately from isolates obtained from other points within the beef production chain. Apart from an isolate from CFeces, that was within 18 SNP of MSewage isolates (=2) (Appendix A), the lowest SNP pairwise differences between CHumans and MSewage vs. all cattle-sourced isolates were 552 and 718, respectively. Minimum span tree (MST) generated with Bionumerics and whole genome MLST (wgMLST) corroborated our phylogenetic analysis and showed distinct clusters of cattle- versus human-sourced isolates (Figure 4; Appendix A). Subsequently, a separate phylogenetic tree which featured only true ESBL isolates also showed limited intermixing between cattle and human isolates (Appendix A). BLAST atlas of isolates identified as non-ESBL versus true ESBL-EC by WGS did not reveal much diversity among non-ESBL and ESBL-EC sequences (Figure 5), although the genetic elements that appeared to be commonly absent in both *E. coli* groups were associated with prophage elements when compared to the reference genome (data not shown).

Using MLST, 58 sequence types were identified for all but five isolates, which did not match any known ST groups. CFeces isolates generally belonged to ST groups 224 (37.5%), 10 (20%), 23 (5%), and unknown ST (5%) (Figure 6; Appendix A). CBasins isolates were mainly identified as members of ST10 (23.8%), 224 (9.5%), and 515 (9.5%). For SStreams isolates, ST69 (14.3%) and ST155 (14.3%) were the most common, followed by STs 244, 515, and 7618 (all at 9.5%). All four isolates from BProcessing belonged to ST20. The pandemic ST group 131 exclusively occurred in MSewage and CHumans, accounting for the largest portion of the isolates at 26.7% and 52%, respectively. In MSewage isolates, other common ST groups included 38, 44, 405, and 1193, while other common ST in CHumans isolates were ST 648 and 38.

In silico typing using the ECTyper and EcOH databases yielded 67/162 and 37/162 unknown and novel O-antigen loci, respectively. About half of the clinical isolates (52%; *n* = 13) were O25:ST131, while three MSewage isolates were of the wzx-Onovel31:ST131 subtype. The majority of cattle-sourced isolates were found to be O8, O89, O9, O15, and O99. Two BProcessing O group isolates remained unclassified, while the other two isolates both classified as O128:H2. All presumptive ESBL-EC identified by MLST as ST224 (*n* = 21) showed an unknown O-antigen with a H23 flagella antigen.

### 3.5. Prevalence of Plasmids and Integrative Conjugative Elements

Most of the presumptive ESBL-EC (98.8%) harbored at least one of the 38 different incompatibility (Inc) and colicinogenic (Col) plasmid types detected (Figure 7; Appendix A). The single most predominant plasmid type across sources was the IncFIB(AP001918) occurring in CFeces (27.5%), CBasin (40.5%), SStreams (19.0%), BProcessing (25%), MSewage (86.7%), and CHumans (64%). Other frequently detected plasmid types along the entire continuum included the IncFII (47.9%), ColRNAI (25.2%), IncFIA (21.5%), and p0111 (19.6%). Prevalence of plasmids Col(MG828), ColRNAI, Col156, IncFIA, IncFIB(AP001918), IncFII, and IncFIC(FII) was higher in human-sourced (CHumans and MSewage) than cattle-sourced isolates (CFeces plus CBasins), while p0111 was higher in cattle isolates (*p* ≤ 0.018; Appendix A). CFeces and MSewage isolates showed the highest plasmid diversity (24/38 each), followed by SStreams (16/38). Plasmids Col(BS512), Col(KPHS6), Col8282, Col(MGD2), IncFII(29), IncFII(pCoo), and IncI2 were unique to human-associated *E. coli*, while plasmids Col(IRGK), ColE10, IncFIA(HI1), IncFII(pHN7A8), IncHI2, IncHI2A, and TrfA were unique to cattle-associated isolates.

The ICEberg database used to describe ICE revealed that all 162 *E. coli* isolates harbored these elements. In fact, twenty-two (7.9%) of all the ICE reported in this study were found in all 162 genomes (Table 2; Appendix A). Most of these elements (ICE*Vch*Ban8, ICE*Val*HN437, ICE*Val*HN396, ICE*Val*A056–2, and ICE*Vfl*Ind1) were members of the SXT/R391 ICE family originally found in *Vibrio* spp. Additionally, a proportion of these ubiquitous ICE have been previously reported in *Pseudomonas* spp. (ICE*6441*, ICE*6440*, PFGI-1, and ICE*Pae*LESB58–1), *Salmonella* spp. (ICE*Sen*Ty2–1, ICE*Sb2*, and ICE*Sb1*), and *Yersinia* spp. (YAPI and ICE*Ye1*). Notably, ICE*Ec2* was the only element that was originally described in *E. coli*. Many of the elements were of unknown function, with a few associated with toxin-antitoxin systems, pathogenicity, and mercuric or antimicrobial resistance.

### 3.6. Prevalence of Virulence Genes

A total of 207 different virulence determinants were identified (Appendix A). Some of the most frequently encountered (70–100%) virulence genes in presumptive ESBL-EC were associated with the components of the type II (T2SS) and III secretion system (T3SS), ferric enterobactin esterase, ferric enterobactin transport ATP-binding protein, enterobactin biosynthesis*,* adhesion, fimbral adhesins, *E. coli* common pilus (ECP), and invasion-related outer membrane protein.

Other virulence genes which play various roles in *E. coli* pathogenicity, including the *pap* fimbral adhesins that were associated with the CFeces, MSewage, and CHuman isolates, with higher frequencies in human-sourced *E. coli*. Fimbrial adhesins *f17* variants and *cdt* were identified in only CFeces and CBasins. The *pic* (8%)*, sat* (52%)*, senB* (28%)*, set* (8%)*, sfa* (4%–12%), and *tpc* (8%) genes were also found in CHumans, but not in cattle-sourced isolates (Appendix A). Hemolysin genes (*hlyA, hlyB, hlyC,* and *hlyD*) were more often associated with CHumans (32%) isolates, but were also present in most other sources [CFeces (2.5%), CBasins (4.8%), SStreams (4.8%), MSewage (6.7%)]).

Overall, 100% of the isolates carried *entB, entC, entF, entS, fepA, fepB, fepC,* and *fes* and 99.4% carried *entA*, *fepD, fepG*, and *ompA*. Only five genes (*espL1, espR1, espX1, espX4,* and *espX5**)* were more prevalent (*p* < 0.0001) in cattle (comprising CBasins and CFeces isolates) than human sources (comprising CHuman and MSewage isolates), while 46 virulence determinants including *aslA, fimA, fyuA, gspL*, *irp12*, *iucABCD*, *kpsMD*, *shuATX*, *ybtAEPQSTUX*, *chuSTUVWY*, *espY1Y2Y3*, *papBCDFGHIJKX*, and *hlyABCD* were dominant in human-sourced isolates (*p* ≤ 0.03; Appendix A).

## 4. Discussion

### 4.1. Antimicrobial Resistance Determinants

The overall ubiquity of β-lactam and tetracycline genes among cattle- and human-sourced isolates across the One Health continuum was expected as both physicians and veterinarians use these classes of antimicrobials to treat humans and beef cattle, respectively [17,57,58]. Tetracycline resistance genes, especially the efflux pumps encoded *tet(A)*, have been detected in high frequencies in generic *E. coli* and ESBL-EC from bovines and humans from United States and China [58,59,60] and from Canadian feedlot environments using metagenomics [61]. However, the ribosomal protection mechanisms conferred by *tet(M)* exclusive to cattle isolates has rarely been reported in fecal samples from beef cattle. Metagenomics studies of manure, soil, and wastewater from dairy and beef production systems [61] within the same region in Canada did not identify *tet(M)*. Since the present study investigated only presumptive ESBL-EC, it is possible that *tet(M)* may be present in the selected ESBL-EC subpopulation of the feedlot environment, likely acquired through horizontal gene transfer (HGT) as proposed by Bryan et al. [58,62]. The predominance of *sul2*, *aph(3’’)-Ib*, *aph(6)-Id,* and *floR* has also been previously observed in cattle- and human-sources [63,64,65]. These resistances seem to arise with AMU and likely reflect the use of aminoglycosides, sulfonamides, quinolones, and/or phenicols for therapy in humans and cattle.

The exclusivity of bleomycin anticancer antimicrobial inactivator *ble* [66] in MSewage ESBL-EC likely reflect the sole use of this antimicrobial in humans and not cattle, as sewage would receive human excrement [67,68]. Although this appears to be the first report of the *ble* gene in municipal sewage in Canada, emerging carbapenemase NDM-1 has been associated with bleomycin resistance in clinical *Enterobacteriaceae* and attributed to selective pressure of bleomycin or bleomycin-like molecules [69].

Quaternary ammonia compounds are often used as disinfectants in processing plants and clinical settings. A number of *qac* subgroup members of small multidrug proteins have been previously identified in *Enterobactericaeae* including *qacE*, *qacEΔ1*, *qacF*, *qacG*, and *qacH*. These genes are usually associated with plasmid-mediated class 1 integrons, which carry a variety of ARGs [70,71]. The *qac* genes are frequently found in combination with genes coding for β-lactams, aminoglycoside, sulfonamide, chloramphenicol, and trimethoprim resistance [4,72]. The regular association of *qac* genes with human isolates in this study is probably due to frequent use of hydrogen peroxide, quaternary ammonium compounds, and sodium hypochlorite-based chemical disinfectants in clinical and household settings [73,74]. Considering that only four isolates from BProcessing were evaluated in this study, the likelihood of detecting *qac* genes was limited even though these disinfectants are often used in meat processing plants. Overall, the generally low occurrence of *QACs* in CFeces, CBasins, and SStreams in contrast to occurrence in human isolates may reflect the rare use of these sanitizers in feedlot environments.

Enrichment was required to isolate ESBL-EC from feedlot-related samples but not for samples from human sources. It may be deduced that ESBL-EC in human-sourced samples were levels of magnitude higher, possibly selected on the basis of clinical illness and/or resistant infections. Variants of *bla*_CTX-M-1,-14,-27_ were only associated with ESBL isolates from humans, a finding which is *c*ompatible with other reports of *bla*_CTX-M-14,-15,-27_ association with the global spread of *bla*_CTX-M_ in human clinical isolates [1,75,76]. Although *bla_CTX-M-15_* was more frequently detected in human- than cattle-related isolates, their occurrence in both cattle- and human-sources may possibly imply transmission between humans and animals [75,77] as ESBL-EC can be zoonotic [78]. In this study, *bla*_SHV_ genes were rather rare in contrast to other Canadian investigations where *bla*_SHV2_ variants have been frequently reported [75,79]. An interesting finding of the current work is that both b*la*_SHV-11 and 12_ were detected in a single isolate from CFeces. *E. coli bla*_SHV_-_12_ was reported in one of the first clinical ESBL-EC cases in a dog with recurrent urinary tract infection in Spain in 2000 [80,81]. These findings implicate that it is possible that even minimal interactions may cause resistance gene transfer within a One Health continuum [81].

We also report what is believed to be the first detection of the recently described *bla*_LAP-2_ class A β-lactamase gene [82] from *E. coli* in Canada. The *LAP-2* gene (originally described in 2007) encodes narrow spectrum β-lactamase resistance. All the isolates from CFeces (*n* = 2) and CBasins (*n* = 1) with *LAP-2* also harbored *qnrS1, bla_TEM_-1*, and *dfrA14*, together with several other ARGs. Co-carriage of *LAP-2* and plasmid mediated *qnrS1* has been previously reported, and *LAP* has been found to be associated with the same gene cluster together with insertion elements and transposons in clinical *E. coli* [83] and *Enterobacter cloacae* [82,84]. The present study found that isolates carrying *LAP*were resistant to 8–10 antimicrobial agents and carried at least five distinct plasmid types, implying a high potential for horizontal gene transfer of MDR.

### 4.2. Genotype-Phenotype Antimicrobial Resistance Concordance

There was a strong agreement (≥93.2%) between clinical resistance phenotypes and genetic determinants across the One Health continuum for all antimicrobials tested. Notably, sensitivity and specificity for quinolone (based on *qnr* and *aac(6’)Ib-cr* genes) were initially low (data not shown). Upon examination of *QRDR* mutations, sensitivity catapulted to 100%, with each quinolone resistant isolate harboring a resistant gene and/or mutation. In contrast, WGS specificity reduced to 1.1%, with a large proportion of quinolone susceptible isolates also possessing a gene and/or point mutations. These differences between WGS sensitivity and specificity based only on resistance genes versus a combination of genes and mutations may be attributed to the possibility that not all *QRDR* mutations confer clinical resistance to quinolones or have yet to be confirmed as causing quinolone resistance. A recent study by Varughese et al. [85] found that the commonly reported *S83L QRDR* mutations did not yield quinolone resistance; although Varughese et al. [85] also noted that coupled with other mutations, S83L conferred quinolone resistance in uropathogenic *E. coli*.

To assess and facilitate adoption of WGS as the gold standard for AMR surveillance, a number of studies investigating the sensitivity and specificity of WGS in *E. coli* [86,87] have reported high agreement between a variety of gene determinants and associated phenotypes. In this study, tetracycline, ampicillin, streptomycin, trimethoprim/sulfamethoxazole, and florfenicol resistance exhibited a high concordance for both sensitivity and specificity. An important aspect of WGS for gene detection is the utilization of already identified sequences as references. It is possible that discrepancies between WGS and phenotypic resistance may result from the presence and expression of novel genes which encode for associated resistance phenotypes. Indeed, previous studies have identified phenotypic resistance which were unaccounted for by known resistance genes (e.g., chloramphenicol, gentamicin, streptomycin, and cefoxitin [24]). It is also possible that WGS sensitivity was impacted by the classification of intermediate resistant and sensitive isolates as resistance phenotype negative. In this study, an appreciable proportion of intermediate ceftazidime resistant (37 out 40 intermediate) isolates were found to harbor genes associated with ceftazidime resistance. McDermott et al. [24] made a similar observation, where intermediate resistant isolates were found to harbor resistance genes. The absence of phenotypic resistance in the presence of resistance genes could also be explained by differences in gene expression among isolates [88].

### 4.3. Insights from E. coli Phylogenetics, MLST and Serogroups

Investigation of the genetic relatedness of presumptive ESBL-EC along the One Health continuum showed generally distinct clades, with evidence of genomes clustering by origin from cattle or human sources, as observed by others [25]. Genetic distinction among ESBL-EC originating from cattle versus human sources along the continuum via phylogenetics implies that isolates are separately adapted to their respective environments and that exchange of isolates between these niches maybe limited [25,89]. For example, ESBL-EC ST131 is reported to be highly human–host specific as it is frequently linked to health-care system exposures rather than environmental sources [90]. It could also be inferred that the segments of the beef production system investigated might not be directly implicated in recent human ESBL-EC infections in this region [25]. Ludden et al. [25] and Salinas et al. [91] also observed distinct clusters between *E. coli* isolates from livestock and humans. However, a study by Mulvey et al. [92] highlighted the possibility of linkage of MDR *E. coli* occurring in cattle and human infections in Canadian hospital on the basis of plasmid *bla_CMY_* A/C replicon fingerprint similarities >90%. In this study, comparative WGS discriminated between isolates originating from cattle and farm environments which were phylogenetically related, as were isolates from CHumans and MSewage. The frequent clustering of CFeces and CBasins ESBL-EC suggested that these isolates were closely related, suggesting that CBasins were catching flow from feedlot pens and reducing the dissemination of ESBL-EC to the broader environment. However, since a number of CFeces and SStreams were intermixed, it can be deduced that some ESBL-EC may be potentially released into the broader environment as water from catch basins is often utilized to irrigate surrounding crops.

Interestingly, different STs prevailed in different sources, even between CBasins and SStreams. Similar to other reports, ST244 were dominant [75] in cattle-sources, while hyperendemic ExPEC ST131 (implicated in bloodstream and urinary tract infection) [76] and, to a lesser degree, STs 38 and 648 were predominant in human-sources. *E. coli* ST648 exhibiting an ESBL phenotype have been reported globally in human patients [7,75]. In concordance with the frequent occurrence of ST131 human sources in this study, ESBL-EC isolated from 11 different Canadian medical centers showed a high occurrence of the clonal complex ST131 producing *CTX_-M-15, -14_*_,_ and *_-27_* from Brampton, Calgary, and Winnipeg, illustrating its propensity to cause nosocomial infections in Canada [93]. Additionally, the overall findings of distinct ST occurrence between sources agrees with other reports [25,75], where ST131 was only found in *E. coli* from dogs and not in farm animals [75]. The unique predominance of specific ST also possibly reflects the spread of clonal lineages or the outcome of selective pressures within the diverse segments of the continuum [94,95].

Serogroups in ESBL-EC from CFeces were mostly unknown O serogroups (50%), with a few O8, O89, and O99, while human-sourced ESBL-EC were mostly O25:H4-ST131. Except for a single instance of O26:H11 and wzx-/wzy-Onovel26:H28, our study did not identify any O157 nor any of the other top 6 non-O157 serogroups (O45, O103, O111, O26, O121, and O145) [96]. O157 and the top 6 non-O157 serogroups cause the majority of Shiga toxigenic *E. coli* (STEC)-associated foodborne infections [96,97]. The infrequent occurrence of these STEC serogroups in ESBL-EC along the continuum could be due to the unspecific nature of the ESBL-EC isolation procedure employed, unlike the specific procedures that are used to isolate STEC, such as direct PCR [98] and sequential immunomagnetic separation [99]. Notwithstanding, the O128 serogroup identified in BProcessing isolates from hides have been linked to serious human infections [97] and may serve as a risk of product contamination during processing. The majority of the O25:ST131 clones from human clinical isolates were CTX-M 15 producers, similar to findings of Aslantaş and Yilmaz [100], who characterized ESBL-EC from Canadian dogs. The specific differences between serogroups along the continuum aligns with MLST and phylogenetic differences in this study and further suggests little to no transmission between cattle and humans.

### 4.4. Mobile Elements: Plasmid and Integrative Conjugative Elements

It is thought that livestock and animal-derived foods are potential sources of ESBL-EC in humans because the same ESBL genes or plasmids have been detected in livestock and farm workers [78,101]. In this study, the greatest plasmid overlaps occurring among all sources of the One Health continuum were IncFIB(AP001918) and IncFII. Johnson et al. [102] noted that, regardless of *E. coli* source, IncFIB was predominant in avian, human, and poultry meat isolates in the US, although plasmid replicons and colicin-related genes differed among *E. coli* sources. Likewise, Rodriguez-Siek et al. [103] also found differences in the actual prevalence of plasmid, virulence, phylogeny, and serogroup traits between human and avian *E. coli*, a finding that is compatible with the differences observed between human and cattle isolates in the present study. In Canada, there is little information regarding the occurrence and frequency of plasmid-mediated transfer of ARGs and virulence determinants among *E. coli* originating from different segments of the One Health continuum. Nevertheless, plasmid-mediated horizontal transfer of multiple ARGs is important for resistance circulation among bacteria in various sources of the continuum [57,92], considering that ~99% of the isolates in our study bore at least one plasmid.

Surprisingly, ICE were found throughout all the genomes sourced from the One Health continuum, even though reports of cattle-related *E. coli*-sourced ICE are rare [15]. Apart from the characteristic set of core genes encoding functions essential for self-transmission and maintenance, ICE are known to often carry cargo genes that impart various fitness and adaptive advantages to hosts [104,105]. Consequently, besides other well-characterized MGEs including plasmids and integrons, ICE may play a crucial role in the evolution and adaptation of *E. coli* to various niches, including those environments subject to the selective pressure of antimicrobials. In this study, genes associated with MDR, metal resistance, and pathogenicity functions were noted for some ICE. The presence of diverse ICE families may also imply exchange of genetic material among different bacterial species. For example, the SXT/R391 family of ICE often in *Vibrio* spp. [106] were present in *E. coli* genomes. Further characterization of ARG-bearing MGEs and their roles in gene exchange via conjugation would shed more light on the extent and frequency at which these elements transfer AMR and virulence genes among *E. coli*.

### 4.5. Occurrence of Virulence Genes

Often, STEC are characterized for their stx virulence determinants due to their role in enterohemorrhagic colitis and hemolytic urea syndrome. The repertoire of virulence factors detected in this study included common genes that encode adhesins essential for biofilm formation, host cell invasion, tissue degradation, and host cell death. For example, the fimbria adhesin *ecpABCDE* that encodes the *E. coli* common pilus essential for early-stage biofilm development and host cell recognition [107] were common in most isolates (91.4%). Likewise, the type 1 fimbriae determinants *fimBCDEFGHI* involved in early stage biofilm formation on host mucosa and abiotic surfaces was also common. The detection of *T3SS* effector protein genes in >70% of isolates in this study demonstrate that all these isolates likely harbored the *E. coli* locus for enterocyte effacement (LEE), the most important pathogenicity island for enterohemorrhagic and enteropathogenic *E. coli* [108].

The exclusive occurrence of *f17* fimbriae genes in pathogenic *E. coli* in CFeces and associated CBasins perhaps reflects point-source dissemination of virulence genes from CFeces to CBasins. The *f17* fimbriae mediates binding to host intestinal microvilli and has been linked to diarrhea and septicemia outbreaks in calves and lambs [109]. Additionally, the higher occurrence of *esp* in cattle-sourced *E. coli* and the higher occurrence of *pap*, *fim*, *hly*, and *iuc* in human-sourced isolates probably reflects the importance of these factors in *E. coli* survival or pathogenicity in their respective hosts. All factors considered, the detection of unique virulence genes specific to niche/environmental hosts may support that virulent ESBL-EC occurring in humans did not likely originate from beef cattle. It may also be deduced that, while some virulence genes are ubiquitous due to their importance for fitness and adaptation to biotic and abiotic conditions, other virulence genes may be unique to isolates based on the source characteristic as well as isolate adaptation [26,110].

## 5. Conclusions

This study used epidemiologically robust study design and sample collection to examine geographically and temporally related presumptive ESBL-EC isolates from beef cattle, water sources, and human clinical samples in southern Alberta. Comparison of the genotypic and phenotypic characteristics of these isolates along a One Health continuum revealed differences in prevalence of similar ARGs, MGEs, and virulence factors together with phylogenetic differences, implying that ESBL-EC originating from beef cattle may not play a significant role in ESBL-EC infections in humans in southern Alberta. The ubiquity of diverse plasmid and ICE families could be indicative of HGT, even within the same niche and warrants further investigation. This study also documents the first detection of the *bla*_LAP-2_ β-lactamase in cattle feces and catch basins and of the *ble* anticancer resistance genes in municipal sewage in Canada. From specificity and sensitivity outcomes, this study noted that WGS promises to be a robust method for AMR investigations, but warrants further validation and improvement in AMR phenotype prediction accuracy. Future investigation of ESBL-EC based on sound epidemiological project designs will continue to add to the body of work described here and may provide useful insights at to the significance of similarities and differences identified within the various segments of the One Health continuum.

## Figures and Tables

**Figure 1 microorganisms-08-00448-f001:**
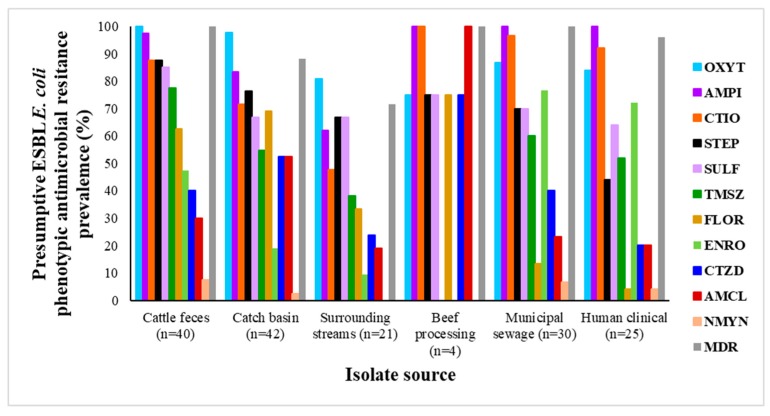
Phenotypic antimicrobial resistance profiles of presumptive extended-spectrum β-lactamase (ESBL) *E. coli* obtained from different sources as described in materials and methods. OXYT—tetracycline; AMPI—ampicillin; CTIO—ceftiofur; STEP—streptomycin; SULF—sulfisoxazole; TMSZ—trimethoprim/sulfamethoxazole; FLOR—florfenicol; ENRO—enrofloxacin; CTZD—ceftazidime; AMCL—amoxicillin/clavulanic acid; NMYN—neomycin; MDR—multidrug resistance. MDR referred to resistance to two or more classes of antibiotics.

**Figure 2 microorganisms-08-00448-f002:**
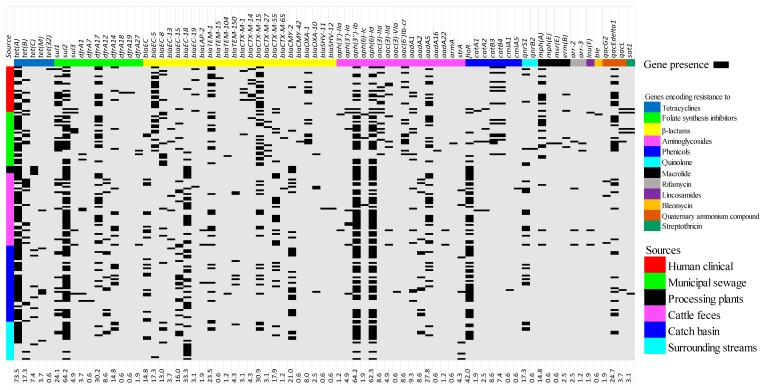
Heat map of resistance genes associated with various classes of antimicrobials as detected by the whole genome sequences of 162 presumptive ESBL-producing *E. coli* across a One Health continuum: Isolates originated from multiple segments of the One Health continuum including human clinical, municipal sewage, beef processing plant, cattle feces, catch basin, and surrounding streams. *aac-(6′)-Ib-cr* classified under aminoglycoside also encodes quinolone resistance. The overall prevalence of each determinant is displayed as a numeric percentage at the bottom of each determinant.

**Figure 3 microorganisms-08-00448-f003:**
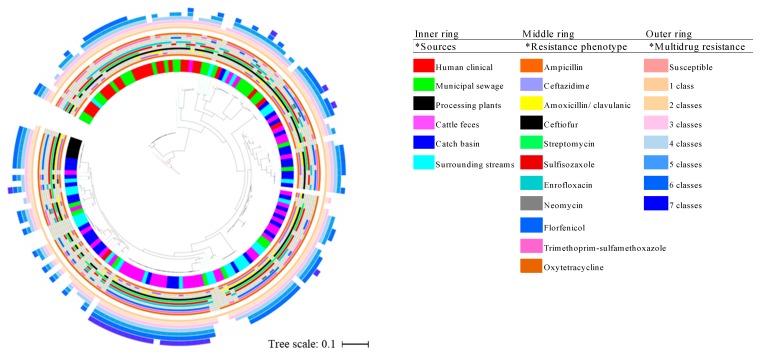
Phylogenetic tree generated on the basis of single-nucleotide polymorphisms (SNPs) of the core genes of 162 *E. coli* isolates obtained from multiple segments of the beef production system and human-associated isolates with reference genome *E. coli* str. K-12 substr. MG1655 (GenBank accession # GI: 545778205/U00096.3). Inner, middle ring, and outer rings are representative of isolate sources, phenotypic antimicrobial, and multidrug resistance, respectively.

**Figure 4 microorganisms-08-00448-f004:**
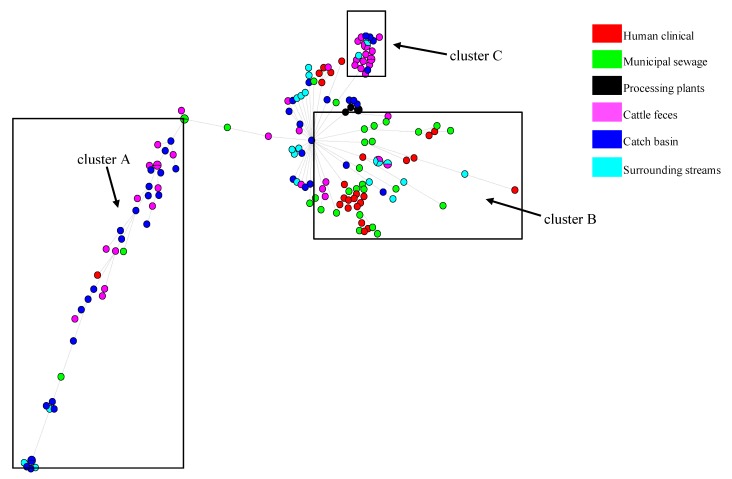
Minimum spanning tree (MST) based on whole-genome multi-locus sequence typing (wgMLST) profiles of 162 presumptive ESBL-producing *E. coli* genomes along the One Health continuum from cattle feces, catch basins, surface streams, beef processing plants, municipal sewage, and humans: The MST includes a total of 9580 wgMLST loci and was constructed with BioNumerics (version 7.6.2). Each circle corresponds to a unique wgMLST profile and is colored based on sample origin. The size of the circle is proportional to the number of isolates sharing the same wgMLST profile, while the branch lengths correspond to the number of allele differences between isolates. Boxes highlight three major clusters of isolates of cattle and human origin.

**Figure 5 microorganisms-08-00448-f005:**
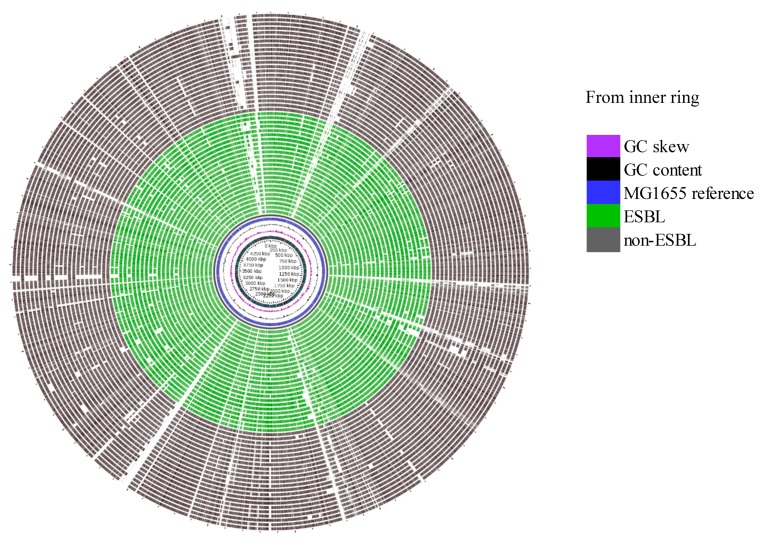
Blast atlas (A—circular) showing sequences similarity and diversity among true ESBL and presumptive non-ESBL *E. coli* using *E. coli* MG1655 as a reference: Isolates shown in this analysis are all cattle-derived (cattle feces, catch basins, surface water, and processing plant) and do not include any human-derived presumptive ESBL-EC because WGS did not identify any non-ESBL-EC from humans or municipal sewage sources based on the presence of *bla*_SHV_, *bla*_TEM_, *bla*_OXA_, or *bla*_CTX-M_. Inner ring comprises GC skew, GC content, and sequence of reference *E. coli* MG1655, while middle and outer rings comprise ESBL and non-ESBL *E. coli*, respectively.

**Figure 6 microorganisms-08-00448-f006:**
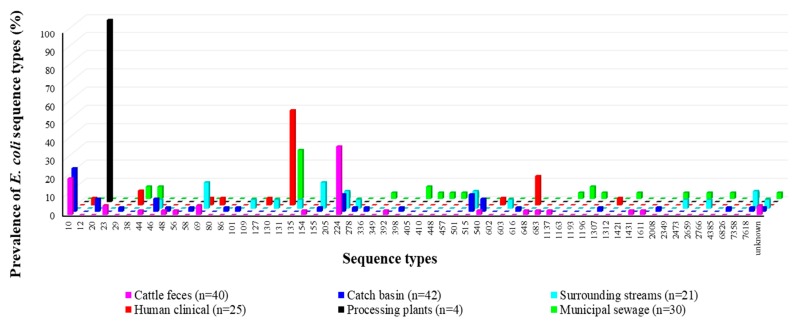
Percentage prevalence of specific sequence types (ST10 to ST7618 and unknown STs) identified in multiple segments of the One Health continuum using the *E. coli* Achtman scheme.

**Figure 7 microorganisms-08-00448-f007:**
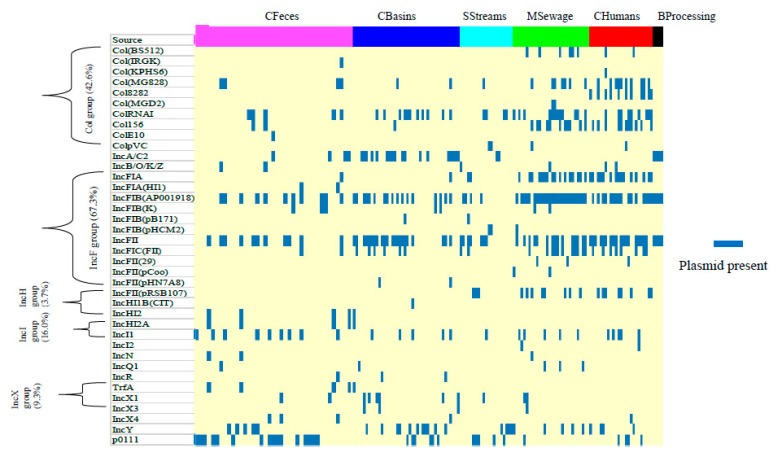
Prevalence of plasmid replicons per source of presumptive ESBL-producing *E. coli* from cattle feces, catch basins, surface streams, municipal sewage, humans, and beef processing along a One Health continuum.

**Table 1 microorganisms-08-00448-t001:** Genotype and phenotype comparison of presumptive ESBL *E. coli* isolated from along a One Health continuum.

	Phenotype: Resistant		Phenotype: Susceptible					
	Gene Positive	Gene Negative	Gene Positive	Gene Negative	Sensitivity (%)	Specificity (%)	PPV (%)	NPV (%)
***Aminoglycoside***								
Streptomycin	111	4	16	31	96.5	66.0	87.4	88.6
Neomycin	8	0	96	58	100.0	37.7	7.7	100.0
***Beta-lactam/beta-lactam inhibitor***								
*Cephems*								
Ceftazidime	61	2	74	25	96.8	25.3	45.2	92.6
Ceftiofor	127	3	8	24	97.7	75.0	94.1	88.9
***Penicillin***								
*Ampicillin*	136	10	0	16	93.2	100.0	100.0	61.5
***Folate pathway inhibitors***								
Trimethoprim/sulfamethoxazole	93	0	29	40	100.0	58.0	76.2	100.0
Sulfisoxazole	112	5	6	39	95.7	86.7	94.9	88.6
***Phenicol***								
Florfenicol	67	3	16	76	95.7	82.6	80.7	96.2
***Quinolones***								
Enrofloxacin	70	0	91	1	100.0	1.1	43.5	100.0
***Tetracyclines***								
Oxytetracycline	144	5	2	11	95.7	84.6	98.6	68.8

Sensitivity (%) equals number of isolates that harbored resistant determinants divided by the total number of isolates exhibiting clinical resistance phenotypes, while specificity (%) was calculated as the number of isolates that did not harbor genetic determinants divided by the total number of phenotypically susceptible isolates. Whole genome sequence of positive predictive value (PPV%) and negative predictive value (NPV%).

**Table 2 microorganisms-08-00448-t002:** Integrative conjugative elements ubiquitously identified in *E. coli* genomes along a One Health continuum.

ICE name	Function	Family	Source Bacteria
ICE*Vch*Ban8	toxin-antitoxin system	SXT/R391	*Vibrio cholerae* MZO-3
HAI2	Unknown (-)	Unclassified	*Erwinia carotovora* subsp. atroseptica SCRI1043
ICE*CroI*CC168–1	Unknown (-)	Unclassified	*Citrobacter rodentium* ICC168
ICE*6441*	Unknown (-)	Unclassified	*Pseudomonas aeruginosa* FFUP PS CB5
ICE*6440*	Unknown (-)	Unclassified	*Pseudomonas aeruginosa* HSV3483
AICEScatt35120	Unknown (-)	Unclassified	*Streptomyces cattleya* DSM
ICE*Val*HN437	Unknown (-)	SXT/R391	*Vibrio alginolyticus* HN437
ICE*Val*HN396	Unknown (-)	SXT/R391	*Vibrio alginolyticus* HN396
ICE*Val*A056–2	Unknown (-)	SXT/R391	*Vibrio alginolyticus* 103826
ICE*Kpn*HS11286–2	Unknown (-)	Unclassified	*Klebsiella pneumoniae* HS11286
Tn*6098*	Unknown (-)	Unclassified	*Lactococcus lactis* subsp. lactis KF147
YAPI	Pathogenicity	ICEYe1	*Yersinia pseudotuberculosis* IP 31758
YAPI	Pathogenicity	ICEYe2	*Yersinia pseudotuberculosis 32777*
ICE*Sen*Ty2–1	Unknown (-)	SPI-7	*Salmonella enterica* subsp. enterica serovar Typhi str. Ty2
PFGI-1	Unknown (-)	PAPI-1	*Pseudomonas fluorescens* Pf-5
ICE*Dze*Ech1591–1	Unknown (-)	ICEKp1	*Dickeya zeae* Ech1591
ICE*Sb2*	Unknown (-)	SPI-7	*Salmonella bongori* 2022/77
ICE*Sb1*	Unknown (-)	SPI-7	*Salmonella bongori* CEIM46082
ICE*Ye1*	Unknown (-)	ICEYe1	*Yersinia enterocolitica* Y69
ICE*Ec2*	Unknown (-)	Unclassified	*Escherichia coli* BEN374
ICE*Pae*LESB58–1	Mercuric resistance	ICE*clc*	*Pseudomonas aeruginosa* LESB58
ICE*Vfl*Ind1	Antibiotic resistance; toxin-antitoxin system	SXT/R391	*Vibrio fluvialis* Ind1

Integrative conjugative elements (ICE) were identified by sequence homology via BLAST against the ICEberg database.

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
