# Peer review of "Whole Genome Sequencing Differentiates Presumptive Extended Spectrum Beta-Lactamase Producing Escherichia coli along Segments of the One Health Continuum"

_microorganisms, 2020, doi:10.3390/microorganisms8030448_

Round 1

Reviewer 1 Report

Adator et al. present a manuscript describing antibiotic susceptibility testing and phylogenetics of extended-spectrum β-lactamase E. coli from whole genome sequencing data of 162 E. coli isolates collected from four beef cattle feces and associated feedlot catch basins, a stream adjacent to one feedlot, a number of sampling sites within a beef processing plant, two(?) municipal sewage plants, and 25 clinical human patients. All samples were collected from within a defined geographic region (southern Alberta), although the size and distribution of sampling sites within that region are unclear. The manuscript is well written and the study design, data collection and data analysis appear appropriate for the study objectives, which were descriptive in nature (lines 71 to 74 in manuscript). The hypothesis being tested was that “ESBL-EC from humans would harbor ARGs, MGEs and virulence factors that were distinct from ESBL-EC isolated from the beef cattle production environment.” The objectives of the study were achieved and data also seemed to support a different apparent hypothesis (e.g. wgMLST ESBL-EC from humans would cluster separately or belong to unique phylogenies from ESBL-EC isolated from the beef cattle production environment.) So the STs from the 2 host populations (beef cattle vs. humans) appear to differ, but from the presentation of results it is not clear to me how much the ARGs, MGEs and virulence factors differ between the 2 host populations. Much of the data analysis is devoted to apparently testing a different hypothesis, that the Sts from humans and cattle will differ. I agree the results support the stated hypothesis, humans harbor ARGs, MGEs and virulence factors that were distinct from ESBL-EC isolated from the beef cattle production environment, however there also appear to be many that are similar, and as you correctly state more work is needed to address how these MGEs shared across the one health continuum influence adaption and evolution of host associated STs. In my opinion, there are current limitations in the methods and results descriptions that should be addressed (see specific comments).

General: The genome sequences should be submitted to a reference database? Please provide a supplementary table of the database accession numbers for each of the 162 sequences.

Specific comments:

Line 74 – This is a sticky one – was this the a priori hypothesis? What humans? Broadly, as written this hypothesis might be assumed to include humans in direct contact with the beef cattle production environment; thus the study design does not completely address this broader hypothesis. The sewage samples are likely a good proxy for the general human population, but it is not clear how the sewage plants may or may not be link to the beef cattle production environment. Also, this is sticky because one should not modify your hypothesis post-hoc to fit the study design. Perhaps the hypothesis statements might be refined or extended to fit your actual design and approach and your intent of what you planned to explore and test with the methods used? Lines 568-570 suggest perhaps the hypothesis was more specific than presented at line 74.

Line 86 – do you mean you cultured from clinical tissue samples collected from patients? In my institution, this would require Human subjects IRB review and approval. Did your study undergo institutional review for human subjects research? Or do you mean isolates collected by Calgary Laboratory Services and some randomly selected isolates were provided?

Line 89 – Beef processing plant samples – are these from swabs taken of the various listed surfaces in the plant? What type of swab? If swabs, how were the swabs handled pre-enrichment? Agitated in a stomacher? Do you have a reference for the plant site sampling methods? Were all the plant samples processed individually for culture or was there pooling of samples? 

Line 91 - 94 – how many isolates were obtained from plating the enrichment broth to the selective agar media? How were isolates picked from the selective agar – reference 32 does not include culture methods and reference 33 provides some hints for cattle feces and wastewater samples, but the details regarding transferring enrichment broth to selective plates for cattle feces are not provided. What volume of enrichment broth was passed to the ceftrioxone-MacConkey plate, and then how many and what percent of isolates were selected from these plates to pass to the ampicillin-tryptic soy agar?

At line 95, “ceftrioxone-MacConkey (1μg/ml)” should likely be “ceftrioxone-MacConkey agar (1μg/ml)”

Line 97 – with modifications of Method 1603 using selective plates as described in ref 33? Did you also use non-selective plates as described in ref 33? So you have total E. coli counts and ESBL-EC counts from these samples? As requested at comment for line 91 and 101, the total number of isolates collected and the proportion passed to WGS should be provided in results, differentiated by sampling source.

Line 101-105, a) how many total presumptive ESBL-EC were originally isolated from each source? What was the size of the original sample population from which the subset of 162 isolates was selected? Does the subset of 162 represent all the colonies picked from the original selective media? Do you have data on how many presumptive colonies were observed per sample? This section describing the subset selection to pass to WGS seems out of place because antimicrobial susceptibility testing (AST) to identify MDR isolates follows this section. Was AST done only on this subset of 162 isolates that were then passed to WGS, or was AST performed on a larger number of isolates? How were MDR isolates identified and selected at this stage prior to conducting AST? From line 175 and figure 1, it appears there were only 12 (7.4%) non-MDR isolates (i.e. phenotypically resistant to only ampicillin at initial selection step?) out of 162 isolates. Was being MDR vs. non-MDR actually used as a selection criteria to pass isolates to WGS?

Line 107 – following on the previous comment, how many isolates were tested for AST and what were the selection criteria for AST testing?

Line 124 – please provide more detail regarding the sequencing methods. Were you multiplexing samples? How many runs? How many samples per run? Target coverage depth?

Line 127 – please include parameter settings for these packages.

Line 129 – revise – there is no section 2.6

Line 158 – 161 – please include a description of any criteria used to select the subset of non-ESBL-EC and true-ESBL-EC submitted to this analysis

Line 164-166 – please provide a table of WGS sequencing results for the 162 isolates (likely a table to include in supplementary data) – include quality parameters for assemblies (e.g. coverage, resulting number of contigs per isolate, etc.)

Fig, 2 – sources key is cut off

Author Response

Adator et al. present a manuscript describing antibiotic susceptibility testing and phylogenetics of extended-spectrum β-lactamase E. coli from whole genome sequencing data of 162 E. coli isolates collected from four beef cattle feces and associated feedlot catch basins, a stream adjacent to one feedlot, a number of sampling sites within a beef processing plant, two(?) municipal sewage plants, and 25 clinical human patients. All samples were collected from within a defined geographic region (southern Alberta), although the size and distribution of sampling sites within that region are unclear. The manuscript is well written and the study design, data collection and data analysis appear appropriate for the study objectives, which were descriptive in nature (lines 71 to 74 in manuscript). The hypothesis being tested was that “ESBL-EC from humans would harbor ARGs, MGEs and virulence factors that were distinct from ESBL-EC isolated from the beef cattle production environment.” The objectives of the study were achieved and data also seemed to support a different apparent hypothesis (e.g. wgMLST ESBL-EC from humans would cluster separately or belong to unique phylogenies from ESBL-EC isolated from the beef cattle production environment.) So the STs from the 2 host populations (beef cattle vs. humans) appear to differ, but from the presentation of results it is not clear to me how much the ARGs, MGEs and virulence factors differ between the 2 host populations. Much of the data analysis is devoted to apparently testing a different hypothesis, that the Sts from humans and cattle will differ. I agree the results support the stated hypothesis, humans harbor ARGs, MGEs and virulence factors that were distinct from ESBL-EC isolated from the beef cattle production environment, however there also appear to be many that are similar, and as you correctly state more work is needed to address how these MGEs shared across the one health continuum influence adaption and evolution of host associated STs. In my opinion, there are current limitations in the methods and results descriptions that should be addressed (see specific comments).

General: The genome sequences should be submitted to a reference database? Please provide a supplementary table of the database accession numbers for each of the 162 sequences.

Response: The sequences have been submitted to Bioproject PRJNA556083

Specific comments:

Line 74 – This is a sticky one – was this the a priori hypothesis? What humans? Broadly, as written this hypothesis might be assumed to include humans in direct contact with the beef cattle production environment; thus the study design does not completely address this broader hypothesis. The sewage samples are likely a good proxy for the general human population, but it is not clear how the sewage plants may or may not be link to the beef cattle production environment. Also, this is sticky because one should not modify your hypothesis post-hoc to fit the study design. Perhaps the hypothesis statements might be refined or extended to fit your actual design and approach and your intent of what you planned to explore and test with the methods used? Lines 568-570 suggest perhaps the hypothesis was more specific than presented at line 74.

Response 1: To eliminate any implied broader hypothesis, hypothesis has been revised (line 75). The sewage treatment plants were located upstream and downstream of the major feedlot cattle production sites in the province as described by Zaheer et al, 2020; sewage samples were collected from Calgary and Medicine Hat and were located in the same regional area as the Calgary Laboratory Services from which human isolates originated (description has been updated in Line 88).

Line 86 – do you mean you cultured from clinical tissue samples collected from patients? In my institution, this would require Human subjects IRB review and approval. Did your study undergo institutional review for human subjects research? Or do you mean isolates collected by Calgary Laboratory Services and some randomly selected isolates were provided?

Response 2: Human clinical presumptive ESBL-EC isolates were randomly selected from anonymous individual patients during the same period.  Complete anonymity resulted in the study not requiring apatient approval or a clinical use permit.

Line 89 – Beef processing plant samples – are these from swabs taken of the various listed surfaces in the plant? What type of swab? If swabs, how were the swabs handled pre-enrichment? Agitated in a stomacher? Do you have a reference for the plant site sampling methods? Were all the plant samples processed individually for culture or was there pooling of samples? 

Response 3: Apart from ground beef and retail meat, remaining samples were obtained from swabs. Sampling procedures followed those described by Gills and Jones 2000.  This reference has been added to the manuscript. Samples were subject to enrichment as outlined in the paragraph immediately following the description of the sample collection methodology.

Line 91 - 94 – how many isolates were obtained from plating the enrichment broth to the selective agar media?  three How were isolates picked from the selective agar – reference 32 does not include culture methods and reference 33 provides some hints for cattle feces and wastewater samples, but the details regarding transferring enrichment broth to selective plates for cattle feces are not provided. What volume of enrichment broth was passed to the ceftrioxone-MacConkey plate – saturated swab was used to inoculate plates, and then how many and what percent of isolates were selected from these plates to pass to the ampicillin-tryptic soy agar? Three randomly selected colonies were banked.

Response 4: Details regarding isolation procedures have been provided.

At line 95, “ceftrioxone-MacConkey (1μg/ml)” should likely be “ceftrioxone-MacConkey agar (1μg/ml)”

Response 5: “ceftriaxone-MacConkey (1μg/ml)” corrected to “ceftriaxone-MacConkey agar (1μg/ml)” Note hat we have also corrected the spelling of “ceftriaxone”.

Line 97 – with modifications of Method 1603 using selective plates as described in ref 33? Did you also use non-selective plates as described in ref 33? So you have total E. coli counts and ESBL-EC counts from these samples? As requested at comment for line 91 and 101, the total number of isolates collected and the proportion passed to WGS should be provided in results, differentiated by sampling source.

Response 5: Yes , selective and non-selective plates were used as described in reference 33. Total E. coli counts, and ESBL-EC counts from these samples were not available as serial dilutions for enumeration were not conducted.  We have indicated that 3 randomly selected colonies from selective and non-selective plates were picked for banking.  These colonies were sub-cultured onto TSA-ampicillin. In a separate study where phenotypic ASTs and PCRs were used, we describe 705 ESBL-EC and 663 generic E. coli used in that study. Subsequently, a subset (n=162) of the 705 presumptive ESBL-EC were selected for sequencing in the current study.

Line 101-105, a) how many total presumptive ESBL-EC were originally isolated from each source? (CFeces, n = 382), catch basins (CBasins, n=137), surrounding streams (SStreams, n = 59), beef processing plants (BProcessing, n = 4), municipal sewage (MSewage; n = 98) and human clinical specimens (CHumans, n = 25).  What was the size of the original sample population from which the subset of 162 isolates was selected? 705 isolates. Does the subset of 162 represent all the colonies picked from the original selective media? No 3 random colonies were selected from the selective plate….only one of these 3 colonies was randomly selected for WGS. Do you have data on how many presumptive colonies were observed per sample? We did not count colonies on each plate only randomly selected 3 colonies. This section describing the subset selection to pass to WGS seems out of place because antimicrobial susceptibility testing (AST) to identify MDR isolates follows this section. We have deleted this section and moved it into the AST section.  Was AST done only on this subset of 162 isolates that were then passed to WGS, or was AST performed on a larger number of isolates? Was completed on 705 presumptive ESBL.  How were MDR isolates identified and selected at this stage prior to conducting AST? From line 175 and figure 1, it appears there were only 12 (7.4%) non-MDR isolates (i.e. phenotypically resistant to only ampicillin at initial selection step?) out of 162 isolates.  This is correct MDR was a major selective criteria.  Was being MDR vs. non-MDR actually used as a selection criteria to pass isolates to WGS? Yes this was part of the selection procedure. 

Response: The 162 isolates were selected from a population of 705 ESBL-EC that were previously subjected to AST (manuscript under review). Phenotypic multi-drug resistance was the primary inclusion criteria for isolate selection, but source as well as time of collection were also used as selection criteria. However, Isolates from sources did not exhibit MDR, but if this source was underrepresented, some isolates were still selected for WGS. 

Line 107 – following on the previous comment, how many isolates were tested for AST and what were the selection criteria for AST testing?

Response: All isolates were previously subjected to AST (in this cae Based on isolate sources and sites, 705 presumptive ESBL-EC isolates were subjected to AST).

Line 124 – please provide more detail regarding the sequencing methods. Were you multiplexing samples? How many runs? How many samples per run? Target coverage depth?

Response: Requested details provided. Multiplexing is a common procedure for running bacterial genomes on Nextera XT/V3 cartridge runs and so we initially felt that this point was a given. Samples were run 16 samples/v3 cartridge (2x300bp);with a 2x300bp paired-end format, which was previously described in line 124.

Line 127 – please include parameter settings for these packages.

Response: All parameters used in settings have been included.

Line 129 – revise – there is no section 2.6

Response: Revised as “section 2.4”

Line 158 – 161 – please include a description of any criteria used to select the subset of non-ESBL-EC and true-ESBL-EC submitted to this analysis

Response: Description has been included in text. The presumptive-ESBL-EC and true-ESBL-EC were randomly selected to represent each source of the beef production system (with consideration that ideal number of genomes for a GView atlas is 50 sequences), plus the fact that all human isolates were identified as true ESBL producers by WGS and thus no presumptive ESBL-EC were included from this group.

Line 164-166 – please provide a table of WGS sequencing results for the 162 isolates (likely a table to include in supplementary data) – include quality parameters for assemblies (e.g. coverage, resulting number of contigs per isolate, etc.)

Response: Assembly statistics have been included in supplementary data Table S3.

Fig, 2 – sources key is cut off

Response: Key has been corrected

Reviewer 2 Report

This is an outstanding manuscript presenting a very complete characterisation of Escherichia coli ESBL isolated from a wide variety of sorces from the one-health continuum. Such studies, not only allow us to better understand the ecology of this particular bacterium but also to demonstrate that the dissemination of antimicrobial resistances is not just a matter of clinical, environmental or animal factors, but a combination of them all.

I just have a few minor issues that I would like the authors to address before the publication of this article.

L26 ceftriaxone. Correct throughout the manuscript.

L74-76 as written, these lines seem a little bit out-of-place from my point of view. Could them be better integrated into the paragraph?

L86 abdominal?

L97-99 These lines should be removed and integrated into the results section.

L101-105 These lines are referred to the isolation and should not be in the introduction. I think they would fit better into the material and methods section.

L114 Give details about how the glycerol stocks were made.

L116 How was the calibration done?

L163-166 These lines should be merged in section 3.4

L170 Why MIC Amp breack-point? If an antibiogram was used, it is better to use synergy of amoxi-clavulanic with 2nd and 3rd gen cephalosporins to identify potential ESBL producers.

L175-176 MDR resistance is based on the resistance to three classes, not two (Clin Mic Infect (2012) 18(3):268-281), modify manuscript accordingly.

L184-233 This section should be subdivided for better clarification, based on the typology of the resistance genes.

L254 (Fig.2) The legend is missing for the light blue colour.

L418-419 delete italic font

L584-602 List of abbreviations should be in alphabetical order. ExPEC and ICE should be included.

Author Response

This is an outstanding manuscript presenting a very complete characterisation of Escherichia coliESBL isolated from a wide variety of sorces from the one-health continuum. Such studies, not only allow us to better understand the ecology of this particular bacterium but also to demonstrate that the dissemination of antimicrobial resistances is not just a matter of clinical, environmental or animal factors, but a combination of them all.

We appreciate the reviewer’s praise of our work.

I just have a few minor issues that I would like the authors to address before the publication of this article.

L26 ceftriaxone. Correct throughout the manuscript.

Response 1: Corrected throughout manuscript we apologize for this obvious oversight.

L74-76 as written, these lines seem a little bit out-of-place from my point of view. Could them be better integrated into the paragraph?

Response 2: Line 74 – 76 have been revised

L86 abdominal?

Response 3: we have deleted description of isolation sites as this was not relevant to our analysis as all isolates were simply grouped as clinical isolates from humans.

L97-99 These lines should be removed and integrated into the results section.

Response 4: These lines can now be located in results – lines 207 -208

L101-105 These lines are referred to the isolation and should not be in the introduction. I think they would fit better into the material and methods section.

Response 5: Introduction lines 73 – 74 has been modified, while martials and methods lines 101 – 105 have been maintained and expanded upon as requested by reviewer 1 and reviewer 2 below.

L114 Give details about how the glycerol stocks were made.

Response: Details given in lines 101 - 102

L116 How was the calibration done?

Response: Information given

L163-166 These lines should be merged in section 3.4

Response: Lines 163-166 have been merged with section 3.4

L170 Why MIC Amp breack-point? If an antibiogram was used, it is better to use synergy of amoxi-clavulanic with 2nd and 3rd gen cephalosporins to identify potential ESBL producers.

Response: MIC was used for initial presumptive ESBL isolation as mentioned in lines 99 – 100. However, subsequent phenotypic characterization for ESBL producers used the AST/ antibiogram approach described in section 2.2 using the ceftazidime/ clavulanic acid (in agreement with reviewer’s suggestion). The statement initially made in L170 was to point out the fact that initial presumptive isolation using MIC versus subsequent characterization using antibiograms may have accounted for the few isolates that were later found to be ampicillin susceptible by the AST approach.

L175-176 MDR resistance is based on the resistance to three classes, not two (Clin Mic Infect (2012) 18(3):268-281), modify manuscript accordingly.

Response: Data has been modified accordingly

L184-233 This section should be subdivided for better clarification, based on the typology of the resistance genes.

Response: The section has been subdivided

L254 (Fig.2) The legend is missing for the light blue colour.

Response: The legend has been corrected

L418-419 delete italic font

Response: Font formatted

L584-602 List of abbreviations should be in alphabetical order. ExPEC and ICE should be included.

Response: List of abbreviations have been reshuffled in alphabetical order and ExPEC and ICE have been included